# Functional perspectives in mental jigsaw puzzles: Insights from eye-tracking, questionnaire, and behavioral data

**Tsuyoshi Yoshioka**[1]*, **Hiroyuki Muto**[2], **Jun Saiki**[1]

**1** Graduate School of Human and Environmental Studies, Kyoto University, Kyoto, Japan, **2** Graduate School of Sustainable System Sciences, Osaka Metropolitan University, Sakai, Osaka, Japan

* yoshioka.tsuyoshi.y65@kyoto-u.jp

## Abstract

This study investigated cognitive strategies in mental jigsaw puzzles, integrating mental rotation and translation with a focus on directionality and detour arguments. Unlike object mental rotation tasks, these puzzles introduced physical constraints, revealing systematic directional tendencies in both eye movements and subjective reports. Specifically, smaller protruding objects were consistently directed toward larger indented objects. This was accompanied by longer completion times and reduced linearity, paralleling strategies used in physical puzzle-solving. Behavioral asymmetries observed in the puzzles unexpectedly mirrored those found in object mental rotation tasks. While controlling for mental motion directions showed comparable completion times at 300° between tasks, the study did not fully clarify the role of detours, indicating the need for further research.

## 1. Introduction

Jigsaw puzzles, along with related activities such as building blocks and digital games like Tetris®, are widely recognized for engaging mental rotation skills, as suggested by various studies [1–4]. Some studies treat rotation tasks as involving both matching and fitting processes [2,5], such as aligning and joining puzzle pieces, and suggest that these processes share similar cognitive demands across related tasks [4,6]. However, task demands may differ significantly [7]. This may be reflected in the mixed outcomes of studies with primary school children, where puzzle-like mental rotation training showed benefits in arithmetic performance [8,9] but not in matching and puzzle-like tasks [10]. This study pioneers the investigation of cognitive strategies in two tasks under voluntary conditions and could offer deeper insights into cognitive processes in real-world contexts, despite the complexity of cause-and-effect relationships.

Controlling experimental variables is fundamental to uncovering clear cause-and-effect relationships, as seen in managing trajectories in mental scanning tasks [11] or imagery [12,13] and directionality [5] in object mental rotation tasks [14]. However, such constraints may not fully reflect the complexity of cognitive strategies in real-life scenarios. For instance, in everyday activities, smaller protruding connectors typically move toward larger indented connectors, as seen when plugging a protruding terminal into an indented power strip. This cognitive strategy persists even though the process can be reversed. While connectors are

**Data availability statement:** Stimuli, raw data, and code are available on the Open Science Framework (OSF) platform at https://osf.io/4ku38/, where preregistration information and preprint versions (originally made available in 2023) are also accessible.

**Funding:** This research was supported by JSPS KAKENHI (https://www.jsps.go.jp/j-grantsinaid/) Grant Number 20H00107 awarded to JS. The funders had no role in study design, data collection and analysis, decision to publish, or preparation of the manuscript.

**Competing interests:** The authors have declared that no competing interests exist.

often described using gendered terminology in electronics, to avoid confusion with biological gender categorization, the present study refers to protruding terminals as *T-type* objects and indented terminals as *U-type* objects (Fig 1A). Reflecting on mental rotation tasks involving pairs of T-type objects, a question arises: *Could cognitive strategies differ if the pairs involved one T-type object and another U-type object?*

The novelty of the present study lies in its task design employing U-type and T-type objects, contrasting with traditional tasks focused solely on matching T-type objects. This study introduces a *mental jigsaw puzzle paradigm* to explore cognitive strategies (Fig 1B and 1C). Unlike conventional object mental rotation tasks, which primarily assess rotational alignment, this task integrates both rotation and translation, incorporating physical constraints that must be considered when fitting objects together. This design allows for a more ecological investigation of cognitive strategies, bridging the gap between theoretical rotation tasks and real-world spatial problem-solving.

Participants were asked to judge whether two objects matched in matching tasks (MT: a pair of T-type objects) or fit together in fitting tasks (FT: a pair consisting of a U-type object and a T-type object). In FT, physical constraints required participants to consider how objects interlock when fitting them together. However, the space in the U-type object corresponds precisely with the T-type object in MT, suggesting that FT could be solved similarly to MT by focusing on the space. This task design offers a new perspective on mental rotation tasks by integrating spatial reasoning with physical constraints.

A study by Mutlu et al. [4] supports this matching approach, where participants were instructed to judge whether a puzzle piece would 'fit' into a gap, referring to this fit as a 'match' condition, indicating similar cognitive processes. Similarly, Frick et al. [15] and Frick and Pichelmann [6] demonstrated the interchangeable use of 'fit' and 'match' in describing the cognitive process of aligning a rotated ghost with a corresponding hole in puzzle-like tasks. Their findings showed a medium correlation between their puzzle-like tasks and object mental rotation tasks involving matching objects [6].

While ignoring physical constraints may be theoretically sound, it might not apply well in real-world contexts. The purpose of jigsaw puzzles extends beyond merely judging a visual match or fit, aiming to achieve a physical fit, as seen in daily activities like fitting suitcases into a car trunk [16]. Motor simulation, involving both visual and physical aspects, prepares for subsequent actions. Therefore, employing strategies similar to those used in physical puzzles might be more effective. This task design allows for detailed investigations of cognitive strategies, particularly in terms of directionality.

## 1.1. Directionality

In this study, *directionality* refers to the preferred orientation or movement direction that entities possess. For example, people tend to prefer the path requiring the smallest degree of rotation during manual rotation tasks with three-dimensional (3D) T-type objects, manipulated via a hand knob, even if a longer path in the opposite direction is possible (e.g., rotating 60° clockwise instead of 300° counterclockwise), as shown by Wohlschläger and Wohlschläger [17]. Directionality encompasses different types of movements, similar to the motion of a rigid body: specifically, translation (leftward/rightward, backward/forward, downward/upward) and rotation (counterclockwise/clockwise around axes: pitch, yaw, and roll). Radial movements (inward/outward) are considered a type of translation, as they involve movement toward or away from a central point.

Xue et al. [5] were among the first to systematically investigate object mental rotation tasks under conditions where translation directions were controlled in advance, demonstrating that cognitive processing patterns vary depending on the specific object being rotated and aligned.

**A** Terminology

Japanese kanji characters

U-type    T-type

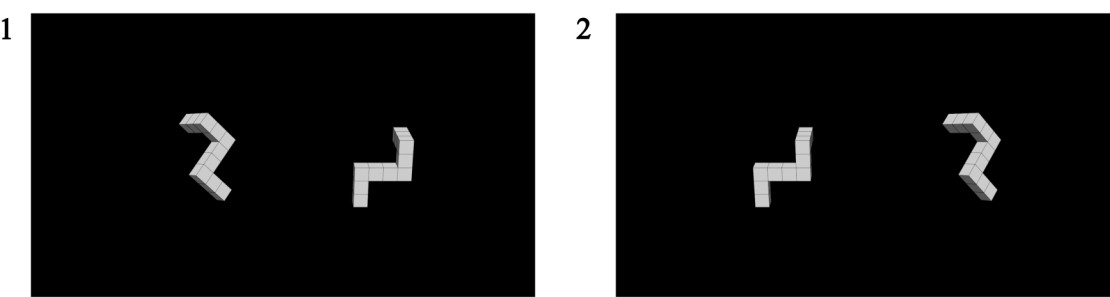

Dip    Bump

**B** Fitting Task (FT); U-type objects are rotated

1    2

**C** Matching Task (MT); T-type objects are rotated

1    2

**Fig 1. Examples of visual stimuli used.** In each stimulus, participants were asked to judge whether two objects matched or fit. **(A)**: The present study refers to objects (connectors) with a 'dip' as U-type and those with a 'bump' as T-type, inspired by the shapes and meanings of the Japanese kanji characters for concave and convex. **(B.1)**, **(C.1)**: objects are rotated 60° on the left side. **(B.2)**, **(C.2)**: objects are rotated 60° on the right side. **(B.1)** represents the less-wall side, while **(B.2)** represents the wall side. The U-type and T-type objects fit together to make a whole cube. The objects to be matched or fit are wholly or partially shaped, respectively, yet they occupy the same space.

This highlights the importance of considering directionality in cognitive tasks, as different strategies can emerge based on these directional preferences. Various outcomes in directionality may arise when the tasks in this study are performed voluntarily.

One possibility is that no specific directionality bias exists. In tasks such as FT and MT, where two objects need to be simply matched or fit together, there may be no specific directional bias. This perspective is consistent with findings from eye-tracking research, such as Just and Carpenter [18], which demonstrated that multiple saccades are exchanged between two objects during matching, indicating no inherent preference for one direction over another.

However, certain contexts may introduce directionality biases. The first potential bias is a left-to-right preference. This might arise due to a common lateral bias, similar to the one observed in reading this sentence, where movement progresses from left to right. Walker [19] demonstrated that people often depict motion in still images with a left-to-right directional bias, suggesting that this preference is deeply rooted in visual processing and may influence how we approach rotation tasks.

In contrast, a right-to-left bias could also emerge, particularly in right-handed individuals. The proximity of objects to the right hand, which is dominant for most people [20], may contribute to this bias. For example, the handedness questionnaire by Nicholls et al. [20] suggests that right-handed individuals often hold a matchbox with their left hand while striking a match with their right. This tendency might encourage a preference for right-to-left directionality in tasks that involve rotation or alignment.

Another bias to consider is a rotated-to-canonical position bias. Tarr and Pinker [21] found that people tend to rotate objects into their canonical positions, or the most familiar orientations. This bias is likely to appear in FT, particularly when rotating a larger U-type object (remnants of a rotated cube) to align with a smaller T-type object to create an upright cube. This process reflects the cognitive preference for recognizing objects in their familiar orientations.

Lastly, a piece-to-assembly bias, commonly observed in jigsaw puzzle solving, may also emerge. Jigsaw puzzles often involve transferring smaller pieces to a larger, partially completed assembly, and this bias may become evident when rotating smaller T-type objects to fit with larger U-type objects.

## 1.2. Trajectory

Beyond directionality, there may be two distinct trajectory strategies in solving FT, as it might be approached similarly to physical jigsaw puzzles. Unlike MT, which lacks obstacles, FT introduces barriers that could require participants to navigate puzzle pieces through indirect routes (detours) rather than direct routes (shortcuts). This study considers multidisciplinary literature to incorporate more ecological perspectives, thereby enriching the review.

(A) **Shortcut**

*Mental Rotation:* Shepard and Metzler [14] demonstrated a linear relationship between angular disparity and reaction time in mental rotation tasks involving 3D T-type objects, suggesting the use of shortcuts across two sides without barriers for aligning two objects. Indeed, Cave et al. [22] found steeper slopes across temporal distances in sequentially cued tasks with prior shape and orientation information, indicating the involvement of more than just attentional shifts. These findings align with the concept of rotation and translation as two independent physical motions in a rigid body.

*Mental Scanning:* Kosslyn et al. [11] found similar linearity in mental scanning tasks, where participants imagined moving a point across a straight path in a bird's eye view, visualizing unimpeded traversal through a bay's barriers. This supports a cognitive *tunnel* effect [23], favoring the use of shortcuts in such scenarios.

*Mental Navigation:* Thorndyke and Hayes-Roth [24] explored mental navigation in a corporate building, showing that both map learners (bird's-eye view) and experienced navigating learners (worm's-eye view) excelled at estimating distances along direct paths onsite, supporting the idea of cognitive tunneling.

(B) **Detour**

*Motor imagery:* Sekiyama [25,26] revealed behavioral asymmetry in hand laterality tasks, highlighting the challenges of imagining physical movements. Echoing these findings,

Tomasino and Gremese [27] suggested the involvement of the bilateral sensorimotor network in processing bodily stimuli and motor imagery. Notably, the mental rotation of hand tools may activate the premotor cortex in the hemisphere opposite the dominant hand, influenced by the object's manipulability [28], which could similarly play a role in handling puzzle pieces.

*Affordances and Embodied Cognition:* Gibson's affordance theory explains that the environment offers certain actions to actors [29], while embodied cognition examines how the body itself or the body's interaction with the environment shapes cognition [30]. Flusberg and Boroditsky [12] demonstrated the role of embodied cognition, showing that mental rotation slows under motor imagery following the physical experience of heavier objects, with similar effects reflected in mental navigation findings [31].

Lockman [32] and Lockman and Adams [33] studied this interplay through the detour paradigm, observing that infants engage with their environment, particularly barriers, to learn about affordances [34]. In contrast, mature adults quickly resort to detours when encountering barriers, often without considering shortcuts—such as when parking a car.

### 1.3. Present study

The primary objective of this research is to investigate how physical constraints influence cognitive strategies in FT under voluntary conditions.

**Conceptualization and Predictions.** Daily observations suggest that when solving physical jigsaw puzzles, there is often a directional preference for moving smaller pieces toward larger assemblies, with detours commonly employed. These interactions shape cognition and reflect the affordances provided by the puzzle environment, where contextual cues influence movement strategies. In the context of FT, the presence of larger U-type objects may alter the functional role of smaller T-type objects, guiding both their directionality and trajectories.

This study employs subjective reports and eye-tracking to capture potential directional biases. Previous research has employed eye-tracking to investigate various strategic differences, including those across angles [18], sex [35–38], congruency [39], spatial ability [36,40], and time-course [18,36].

In addition to directionality, this study investigates behavioral performance in FT. As shown in Fig 1, imagining the physical fitting of puzzle pieces in condition (B.1) is less complex than in (B.2), as there are fewer obstacles along the way. Furthermore, even within conditions (B.1) and (B.2), the degree of obstacles varies depending on the angle (e.g., 60° versus 300°). Cave et al. [22] suggested that longer distances lead to longer reaction times in sequentially cued tasks, involving both mental rotation and translation. Echoing this, longer reaction times in the present study may correspond to longer trajectory lengths due to the presence of obstacles, while shorter trajectories result in shorter reaction times. In contrast, conditions (C.1) and (C.2) involve no obstacles, indicating typical behavioral symmetry around 180°. In sum, examining the relationships between reaction time, angles, and the side of the U-type object may help distinguish between shortcut and detour strategies.

This study took the stance that there will be a piece-to-assembly bias in the directionality argument and a tendency toward detours in the trajectory argument. The primary predictions are as follows: **(1)** an emphasis on the larger U-type object as the destination for the smaller T-type object (functional differences in cognitive strategies), **(2–1)** longer completion times when navigating obstacles (behavioral asymmetry in FT), and **(2–2)** longer times and distinct differences, contrasting with the symmetry and linearity typically assumed in MT (behavioral differences between FT versus MT).

## 2. Materials and methods

### 2.1. Participants

A total of 30 out of 40 participants, all of whom were undergraduate and graduate students with normal or corrected-to-normal vision, and most of whom were right-handed (as assessed by the FLANDERS handedness questionnaire [20,41]), completed the entire main experiment (Table 1). The participation rate aligns with similar eye-tracking studies [37–39] and exceeds the recommended sample size of 25 suggested in a pilot study [7], employing similar methodology to previous research [17,42]. The remaining 10 participants exited early due to reasons such as drowsiness, potential overtime, fluctuations in eye-movement signals, eye fatigue, and dizziness. Of the original 40 participants, 37 completed a post-experimental questionnaire, with seven experiencing the experiment partially. Participants enrolled in the experiment between June 26, 2023, and August 2, 2023.

Ethics approval for this study was provided by the Ethics Committee of the Graduate School of Human and Environmental Studies of Kyoto University (Initial Approval No. 23-H-5; Revised Approval No. 23-H-8). All participants provided written informed consent prior to the commencement of the experiment. This study was conducted in accordance with the Declaration of Helsinki.

### 2.2. Stimuli

Stimuli were created using Blender® version 3.1. Fig 2 presents the bisectional rotation axes, with initial angles at 0°, and objects rotated counterclockwise around the bisectional axis. A total of 576 images were generated, with variables including six angles (0° to 300°), two head types (left and right forward protrusions), same-mirrored pairs (congruent and incongruent), six object types, and two tasks. The T-type object was created by subtracting 115 unit-cubes from a whole cube of 125 unit-cubes, and the U-type object was created by subtracting the T-type object from the whole cube. The visual angles of the objects roughly ranged from 3.9° to 5.5°.

### 2.3. Apparatus

The experiment was conducted in a private dark room with no ambient lighting to minimize distractions. An EyeLink® 1000 Plus Desktop Mount eye tracker was used in binocular remote mode, positioned on a table with a chin rest between the participant and a 15.6-inch monitor, which served as the primary light source, at a distance of approximately 735 mm. A foot pedal with two steps was placed under the table to record behavioral responses. The eye tracker used a 25 mm camera lens with a sampling rate of 1,000 Hz, managed by the supplier's supplemental computer and operating system version 5.50. Only the left eye was tracked to avoid reflections from the monitor on the right eyeglass lens. Eye-tracking data were extracted using the supplier's Data Viewer software version 4.3.1, and the experiment was run using PsychoPy® version 2023.1.2 [43] on a Windows® 10 computer.

**Table 1. Demographics of study participants.**

| Participants | Gender | | | Handedness | | | | Age (years) | |
|---|---|---|---|---|---|---|---|---|---|
| Experienced | Total | Female | Male | NR | Left | Mix | Right | Mean | SD |
| Entire | 30 | 14 | 16 | 0 | 2 | 2 | 26 | 21.1 | 2.0 |
| Entire + Partial | 37 | 16 | 21 | 0 | 2 | 2 | 33 | 21.5 | 2.1 |

A total of 40 participants were recruited. Participants self-reported gender; NR = non-response.

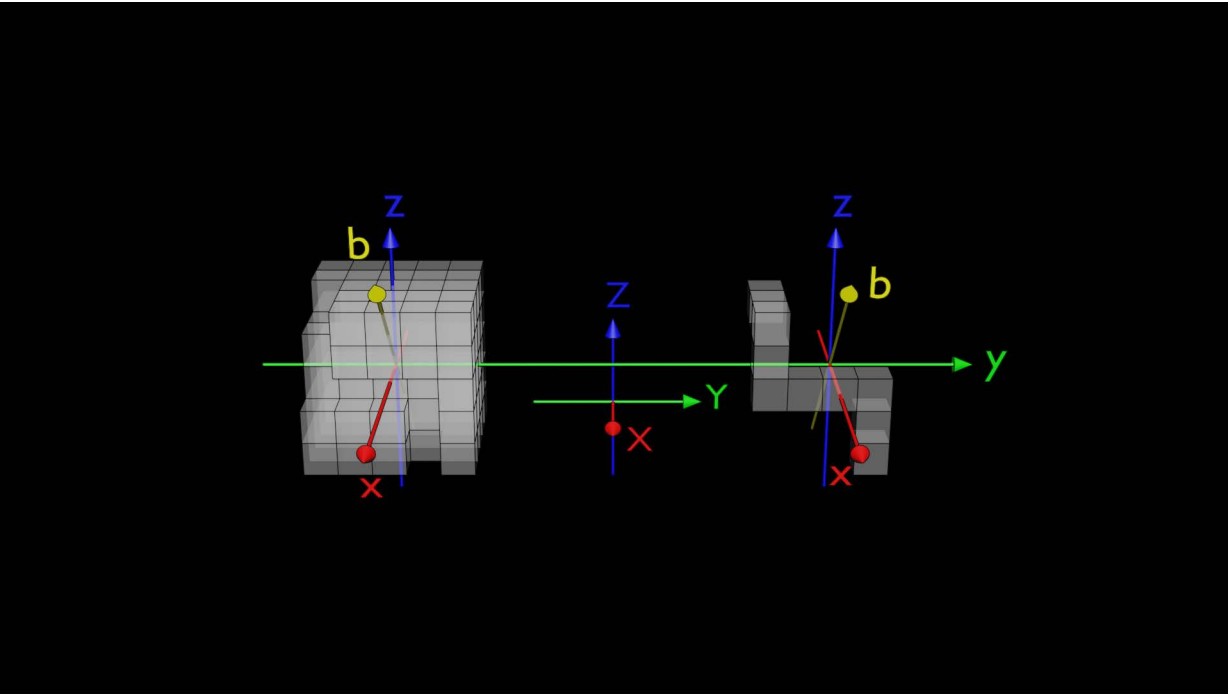

**Fig 2. Bisectional rotation axes in the study.** The yellow axes are bisectional between the local blue and red axes. The cross points of the local axes on both sides correspond to the geometric centers of whole cubes of $5 \times 5 \times 5$ unit-cubes. The directions of the axial arrows and their rotations are aligned with the right-hand rule, where the thumb points in the direction of the bisectional axis, and the curled fingers indicate the rotational direction.

## 2.4. Procedure

Fig 3A outlines the experiment, which included the handedness questionnaire, a practice block, the main block, and a post-experimental questionnaire. The practice and main blocks began with calibrating and validating the eye-tracking (C/V). The entire procedure took approximately 75 minutes. This study used a five-point C/V [39], with validation criteria set to an average error below 0.5° and a maximum error of 1.0°.

Fig 3B details each trial. Participants were instructed to press the right foot pedal if the two objects matched or fit, and the left pedal if they did not, responding as quickly and accurately as possible. Between trials, they were asked to fixate on a cross at the center of the screen. To minimize hand gestures [42] or movements during the experiment, participants were instructed to place their fingers crossed (hands covered) below their stomachs.

The post-experimental questionnaire used a two-alternative forced-choice method with four randomized questions. Participants were asked to indicate whether they generally matched (fit) the left block with the right block or vice versa. Participants were instructed to respond honestly.

## 2.5. Design

This study employed a within-participants design, manipulating angles (Angle: 0°, 60°, 120°, 180°, 240°, 300°), rotation side (R-side: left, right), and task type (Task: FT, MT). Subjective reporting variables included direction (Direction: left → right, left ← right). This research collapsed both head and object types, and analyzed only the congruent condition. Key metrics

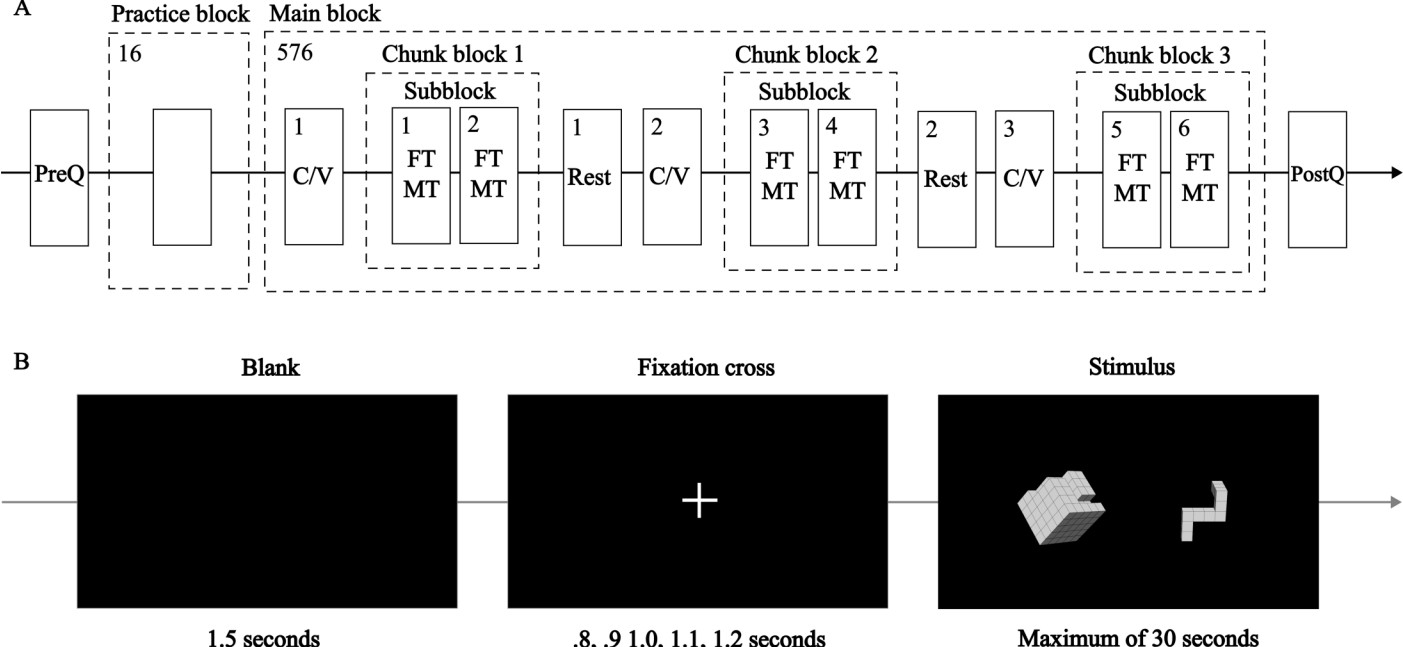

**Fig 3. Schematic of (A) the experiment workflow and (B) a single experiment trial. (A)** PreQ: handedness questionnaire, C/V: calibration and validation of eye-tracking, PostQ: tactics questionnaire. The fitting task (FT) and the matching task (MT) are randomly presented in every trial. The practice block included 16 trials without feedback, using unique stimuli to simulate the main block. The main block consisted of six subblocks, each containing 96 visual stimuli. A total of 576 stimuli were randomly assigned to each subblock, ensuring an equal distribution of conditions (task, congruency, rotation side, angle factors) across the first and last three subblocks. Questionnaires were completed on a tablet computer using custom applications developed in Android Studio® version 2022.2.1. **(B)** A fixation cross appeared randomly, lasting between 800 and 1,200 msec in increments of 100 msec, as a precaution against rhythmical responses. The size of the fixation cross differs from that used in the main experiment. Reaction time and eye-tracking data were measured at the third step.

included reaction time (RT) and error ratio (ER) for behavioral data, dwell time ratio (DTR) for eye-tracking data, and positive responses for self-report data.

## 2.6. Data curation

This study calculated ER as the ratio of errors to total responses, and determined DTR as the ratio of fixation time within one rectangular area of interest (AOI) on the rotation side, to the combined fixation time within two rectangular AOIs. A DTR value greater than 0.5 indicates a bias toward the rotated object's side. Fig 4 provides a detailed example of how the eye-tracking data were calculated. Behavioral and eye-tracking data were collected at the third step shown in Fig 3B.

Behavioral data underwent screening to exclude responses over 30 seconds, those below 200 msec [44], and outliers identified using the median absolute deviation method with a threshold of 2.5 [45]. Approximately 80% of congruent trials were analyzed for mean correct RT, with all participants achieving a trial accuracy rate above 70%.

Eye-tracking data underwent screening to exclude the first fixation, trials without effective records, and fixations outside AOIs, leading to the removal of about 2% of congruent trials. The analysis also excluded trials that were removed from correct RT. Self-report data focused on participants who completed or partially completed the main block and the post-experimental questionnaire. A secondary analysis found no significant gender differences in key metrics, such as RT, ER, and DTR [46]. These findings indicate that gender did not play a significant role in shaping participants' behavioral or eye-tracking patterns in this study. Hence, gender was collapsed in the analysis to streamline the interpretation of results.

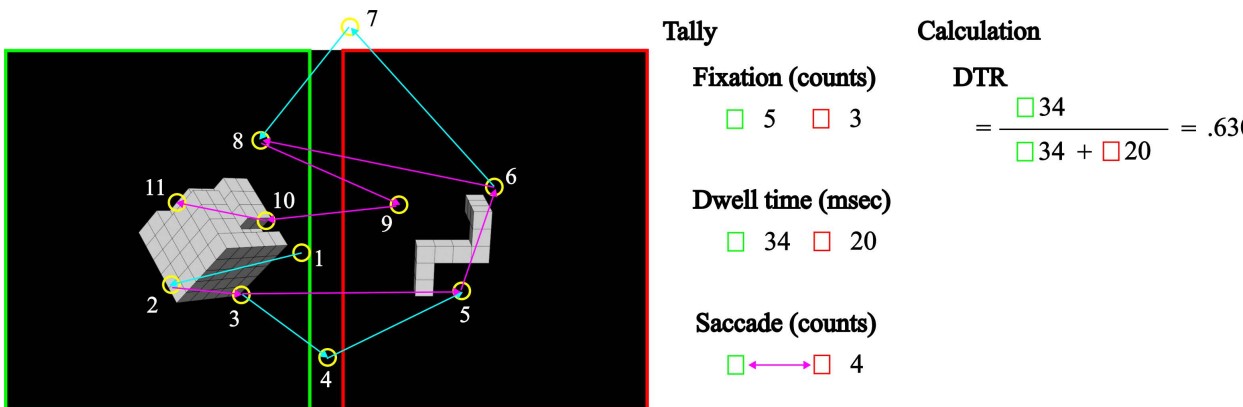

**Fig 4. Schematic of eye-tracking analysis in a single trial.** Green rectangle: left AOI (area of interest), red rectangle: right AOI, yellow circle: fixation, magenta arrow: saccade, cyan arrow: the first saccade and transit saccade, and numerals: fixation sequence and dwell time. DTR: dwell time ratio. The fixation times within each AOI were tallied to calculate the DTR, which is the ratio of fixation time within the AOI on the rotation side to the total fixation time across both AOIs. A DTR value greater than 0.5 indicates a bias towards the rotated object's side. The AOIs were standardized to capture the activity related to two objects located on the left and right sides of the screen, each sharing the same area (960 × 1,080 pixels). The fixations shown are illustrative.

## 2.7. Data analysis

All analyses and visualization were performed using R software [47], along with the following packages: rstatix (version 0.7.2) [48], ARTool (version 0.11.1) [49,50], DescTools (version 0.99.49) [51], ggpubr (version 0.6.0) [52], and gridExtra (version 2.3) [53]. Parametric tests included repeated measures analysis of variance (RM-ANOVA), with Greenhouse–Geisser correction and effect sizes [54], as well as the one-sample Student t-test. Nonparametric tests included RM-aligned rank transform-ANOVA (RM-ART-ANOVA) with effect sizes [50,55], ART contrasts (ART-C) contrast test [56], the one-sample sign test, the Breslow–Day test, the Pearson chi-square test with Yates correction, and the binomial test. All tests were two-tailed, with priority-based and Bonferroni–Holm p-value corrections applied where appropriate.

## 2.8. Detailed explanations on predictions

This study investigates directionality using both eye-tracking and self-report data. If FT is not solved like physical jigsaw puzzles, no bias is expected in DTR, and the self-report data should reflect chance levels. On the other hand, if FT follows physical operations, participants are expected to operate from the smaller T-type object to the larger U-type object, with increased focus on the U-type assembly as the end point of the movement, showing differences from MT.

Additionally, this study analyzes the relationship between reaction times and angle conditions to examine trajectories. If FT is not solved like physical jigsaw puzzles, typical behavioral symmetry, as assumed in MT, should be observed. In contrast, if FT follows physical operations, behavioral asymmetry is anticipated. Specifically, longer reaction times are expected when the U-type object is rotated at 240° and 300° on the left side and at 60° and 120° on the right, while shorter times are expected at 60° and 120° on the left and at 240° and 300° on the right. This pattern would create behavioral asymmetry around 180°, distinct from the symmetrical patterns typically assumed in MT. Furthermore, these behaviors would reflect reduced linearity in FT compared with MT.

## 3. Results

### 3.1. Eye movements

*DTR.* Fig 5A illustrates task differences for DTR. The three-way RM-ANOVA for Task showed a significant interaction effect among Angle, R-side, and Task [$F(5, 145) = 2.842$, $p = .036$, $\eta_G^2 = .008$, $\eta_P^2 = .088$]. A simple effects analysis of the two-way RM-ANOVA for Task showed significant interaction effects of Angle and Task on both rotation sides (all $p < .001$, $\eta_G^2 \geq .026$, $\eta_P^2 \geq .143$). A simple-simple effects analysis of the one-way RM-ANOVA for Task on both rotation sides showed significant main effects of Task at all angles (all $p < .01$, $\eta_G^2 \geq .191$, $\eta_P^2 \geq .338$). The one-sample Student t-test, with Angle and R-side collapsed, showed a significant difference from 0.5 in FT [$t(29) = 16.909$, $p < .001$, $r^2 = .908$]. Hence, the fovea dwelled more on the side with a rotated object, particularly on the side with a rotated U-type object compared to a rotated T-type object. Table 2 provides descriptive statistics for DTR measures.

### 3.2. Subjective report

Fig 5B illustrates task differences for direction proportion. The Breslow–Day test showed a significant non-homogeneity of odds ratios between Direction and Task on the strata of rotation sides [$\chi^2 (1, N = 37) = 44.209$, $p < .001$, W = 1.195]. The post hoc Pearson chi-squared test showed significant associations between Direction and Task on both rotation sides (all $p < .001$, W $\geq .334$).

At individual task levels, the chi-squared test for both FT and MT showed significant associations between Direction and R-side (all $p < .05$, W $\geq .126$). The binomial test for FT showed significant biases from 0.5 on both rotation sides (all $p < .001$); for MT, there was only a significant bias on the right rotation side ($p = .005$). Hence, FT operated in a systematically

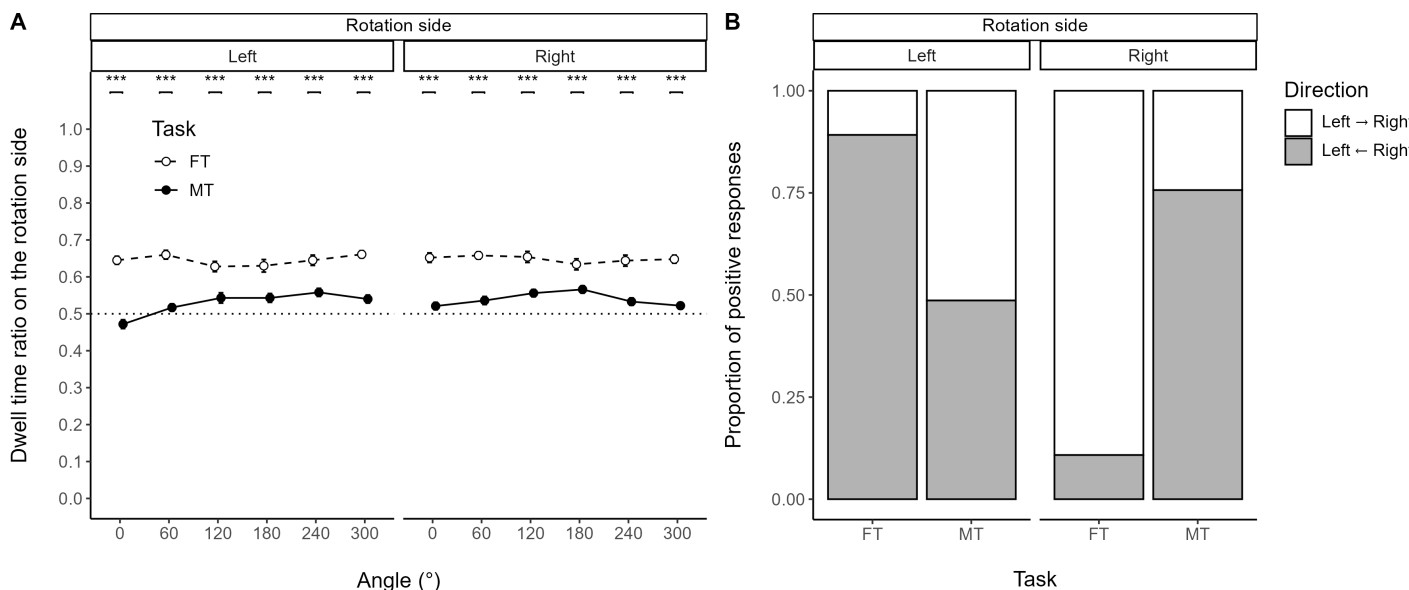

**Fig 5. Functional differences in cognitive strategies. Task differences at rotation-side levels (A) for dwell time ratio across angles and (B) for the proportion of positive responses in subjective matching/fitting direction. (A)** Eye-tracking dwell time ratio (DTR) is the ratio of dwell time within an AOI on the rotation side to that within both AOIs combined. Values above 0.5 indicate more time spent fixating on the rotation side, associated with rotated U-type objects in the fitting task (FT) and T-type objects in the matching task (MT). **(B)** Questionnaire results show significant associations between Direction and Task on both rotation sides. The present study refers to indented object as 'U-type' and protruding object as 'T-type.' In FT, participants tended to match or fit T-type objects leftward toward U-type objects rotated on the left, and rightward toward those rotated on the right. ns: $p \geq .05$, $^*p < .05$, $^{**}p < .01$, $^{***}p < .001$.

**Table 2. Descriptive statistics of dwell time ratio (DTR).**

| Task | FT | | MT | |
|---|---|---|---|---|
| Angle/ R-side | Left | Right | Left | Right |
| 0° | .645 (.060) | .652 (.069) | .472 (.063) | .521 (.053) |
| 60° | .660 (.066) | .658 (.056) | .517 (.054) | .536 (.061) |
| 120° | .628 (.077) | .654 (.084) | .543 (.075) | .556 (.057) |
| 180° | .630 (.092) | .634 (.085) | .543 (.067) | .566 (.055) |
| 240° | .645 (.077) | .644 (.084) | .558 (.058) | .533 (.053) |
| 300° | .661 (.057) | .648 (.062) | .540 (.062) | .522 (.050) |

Values are described as Mean (SD). FT: fitting task, MT: matching task, R-side: rotation side.

directional manner, while MT was systematic on the right but not the left rotation side. Table 3 provides descriptive statistics for subject report measures.

## 3.3. Behavioral asymmetry

Figs 6A and 6B illustrates rotation-side differences and angle-pair differences for correct RT, respectively. The three-way RM-ANOVA for Angle and R-side showed no significant interaction effect among Angle, R-side, and Task ( $\eta_G^2$ =.001, $\eta_P^2$ =.024), but a significant interaction effect between Angle and R-side [$F(3.26, 94.60) = 21.290$, $p < .001$, $\eta_G^2$ =.028, $\eta_P^2$ =.423]. A simple effects analysis of the one-way RM-ANOVA for R-side, after collapsing tasks, showed significant main effects of R-side at three angles (60°, 240°, 300°; all $p < .001$, $\eta_G^2 \geq .050$, $\eta_P^2 \geq .512$) but not at other angles (all $\eta_G^2 \leq .009$, $\eta_P^2 \leq .170$). Another simple effects analysis of the one-way RM-ANOVA for Angle showed significant main effects of Angle on both rotation sides (all $p < .001$, $\eta_G^2 \geq .392$, $\eta_P^2 \geq .686$). The post hoc paired Student t-test on both rotation sides showed significant differences both between 60° and 300° and between 120° and 240° (all $p < .05$, $r^2 \geq .160$).

To explore symmetry after collapsing asymmetric behaviors, the R-side factor was collapsed. The one-way RM-ANOVA for Angle in both tasks, after collapsing rotation sides, showed significant main effects of Angle (all $p < .001$, $\eta_G^2 \geq .321$, $\eta_P^2 \geq .585$), and the post hoc paired Student t-test in both tasks showed no significant difference either between 60° and 300° or 120° and 240° (all $r^2 \leq .081$). Fig 6C illustrates this. Both FT and MT exhibited the same behavioral asymmetry, with the same characteristics at 120°, which diminished when collapsing rotation sides.

## 3.4. Linearity

A previous study by Shepard and Metzler [14] simplified rotation sides to a general direction and reduced angles to a 0°–180° range, facilitating linear regression analysis and allowing for exploratory examination of cognitive shortcuts and detours in FT.

**Table 3. Descriptive statistics of subjective report.**

| Task | FT | | MT | |
|---|---|---|---|---|
| Direction/ R-side | Left | Right | Left | Right |
| Left → Right | 4 | 33 | 19 | 9 |
| Left ← Right | 33 | 4 | 18 | 28 |

FT: fitting task, MT: matching task, R-side: rotation side.

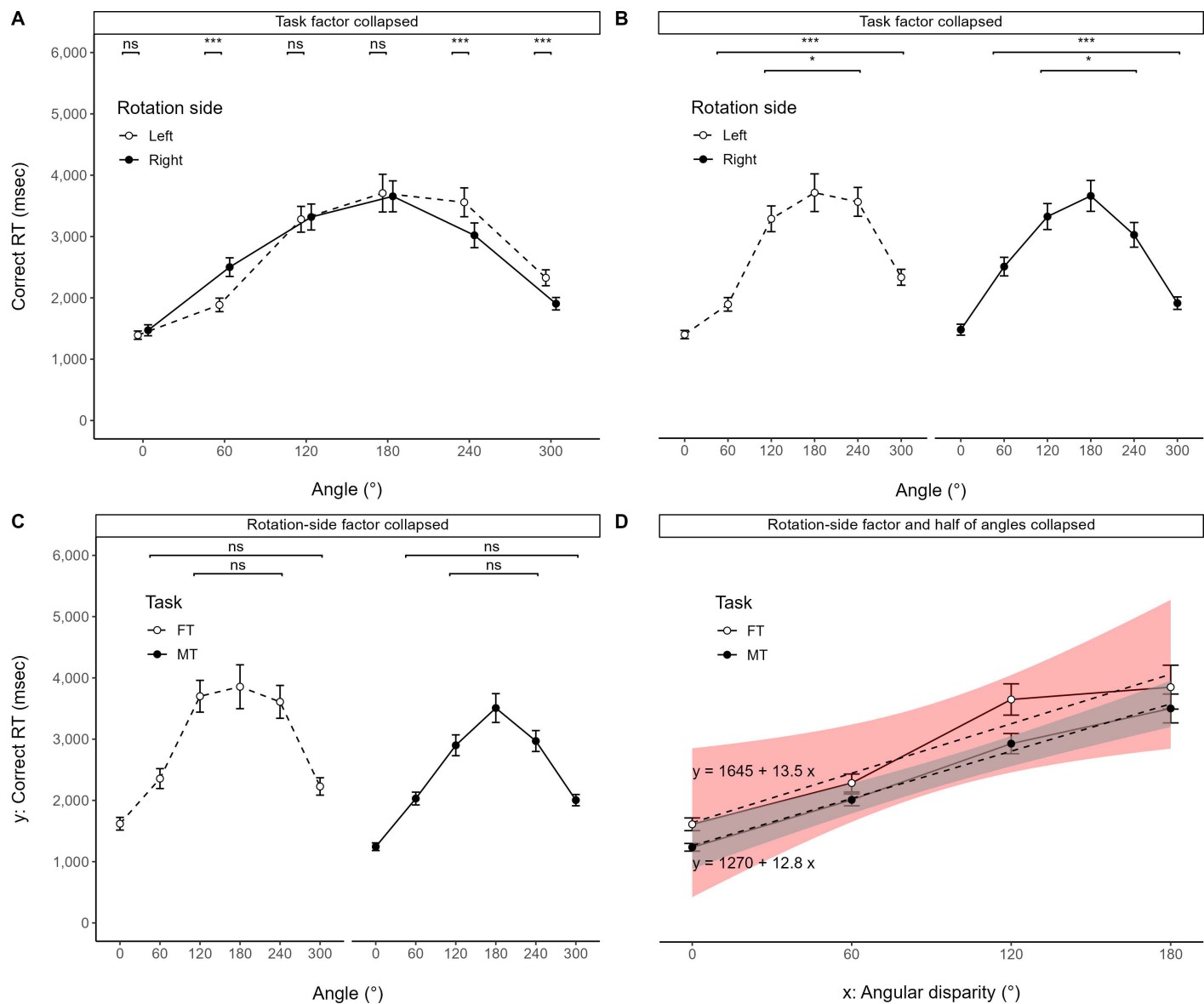

**Fig 6. Behavioral asymmetry. (A)** Rotation-side differences for correct RT across angles after collapsing tasks. Correct RT across angles **(B)** at rotation-side levels after collapsing tasks and **(C)** at task levels after collapsing rotation sides. **(D)** Regression lines for correct RT in tasks across half-round angular disparity after collapsing rotation sides and half of angles. **(B)** Rearranged graph from **(A)**. **(D)** Significant linear trends observed in the fitting task (FT) and the matching task (MT). Dashed lines indicate regression lines with equations; shaded areas represent 95% confidence intervals. ns: *p* ≥ .05, *p* < .05, **p* < .01, ***p* < .001.

A trend analysis, after collapsing to a 0°–180° range, showed significant linear relationships between angular disparity and correct RT for both FT [$F(1, 87) = 126.01$, $p < .001$, $\eta^2 = .325$, $\eta_P^2 = .592$] and MT [$F(1, 87) = 281.64$, $p < .001$, $\eta^2 = .515$, $\eta_P^2 = .764$]. While FT demonstrated significant linearity, the angular disparity explained approximately 19% less variability in correct RT compared to MT. Fig 6D illustrates this trend.

### 3.5. Behavioral difference

Fig 7A illustrates task differences for correct RT. The separate three-way RM-ANOVA for Task showed a significant main effect of Task [$F(1, 29) = 33.338$, $p < .001$, $\eta_G^2 = .040$, $\eta_P^2 = .535$], but

no other significant effects ( $\eta_G^2 \leq .008$, $\eta_P^2 \leq .080$ ). Hence, FT generally took more time overall than MT. Table 4 provides descriptive statistics for correct RT measures.

Task differences in ER were less distinctive, with speed compatible with accuracy. The three-way RM-ART-ANOVA for Task showed no significant interaction effect between Angle, R-side, and Task ( $\eta_P^2 = .017$ ), but a significant interaction effect between Angle and Task [$F(5, 667) = 3.657$, $p = .009$, $\eta_P^2 = .027$]. The post hoc ART-C showed a significant bias of FT over MT only at 120° [$t(667) = 2.680$, $p = .045$]. Table 5 provides descriptive statistics for ER measures.

From the participants' subjective reports (Fig 5B), FT on the left rotation side shared the same direction as MT on the right rotation side. At the same time, rotation also has a direction. This study accounted for these two confounders in an exploratory analysis by controlling for rotational direction and flipping the MT graph on the R-side right level at 180° (Fig 8).

Fig 7B illustrates task differences for correct RT after controlling for direction confounders. The two-way RM-ANOVA for Task showed a significant interaction effect between Angle and

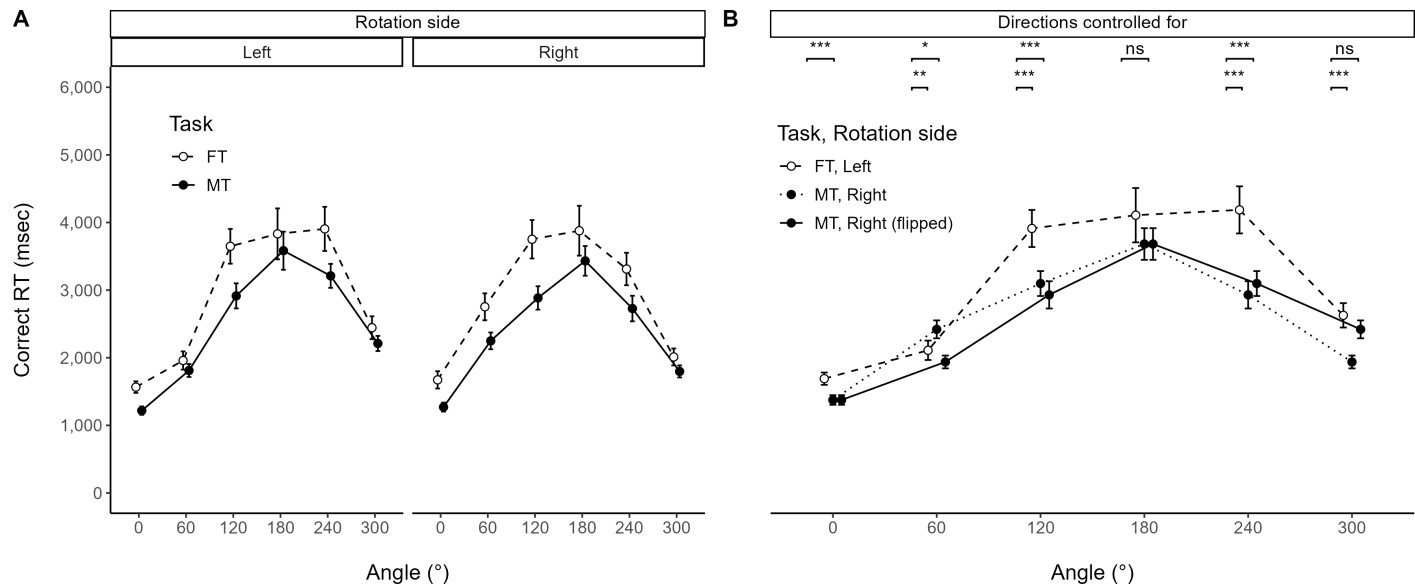

**Fig 7. Behavioral differences between FT versus MT. Task differences for correct RT across angles at rotation-side levels (A) before and (B) after controlling for direction confounders. (A)** Significant main effect of Task observed. **(B)** Controlling for confounders: subjective matching/fitting direction (first and second levels) and both this direction and rotational direction (first and third levels). FT: fitting task, MT: matching task. ns: **$p \geq .05$, *$p < .05$, **$p < .01$, ***$p < .001$.

**Table 4. Descriptive statistics of correct RT.**

| Task | FT | | MT | |
|---|---|---|---|---|
| Angle/ R-side | Left | Right | Left | Right |
| 0° | 1,566 (467) | 1,673 (700) | 1,218 (341) | 1,270 (365) |
| 60° | 1,959 (738) | 2,753 (1,090) | 1,811 (522) | 2,249 (680) |
| 120° | 3,648 (1,407) | 3,752 (1,557) | 2,915 (1,016) | 2,885 (947) |
| 180° | 3,832 (2,064) | 3,879 (2,019) | 3,583 (1,542) | 3,433 (1,201) |
| 240° | 3,905 (1,788) | 3,312 (1,315) | 3,211 (972) | 2,727 (1,034) |
| 300° | 2,444 (926) | 2,012 (687) | 2,211 (613) | 1,797 (488) |

Values are described as Mean (SD). The units are in msec. FT: fitting task, MT: matching task, R-side: rotation side.

**Table 5. Descriptive statistics of error ratio (ER).**

| Task | FT | | MT | |
|---|---|---|---|---|
| Angle/ R-side | Left | Right | Left | Right |
| 0° | 0 (0) | 0 (0) | 0 (0) | 0 (0) |
| 60° | 0 (0) | .083 (.083) | 0 (0) | 0 (.083) |
| 120° | .167 (.167) | .208 (.313) | .167 (.312) | .083 (.250) |
| 180° | .333 (.333) | .250 (.417) | .250 (.313) | .167 (.250) |
| 240° | .167 (.250) | .083 (.333) | .083 (.229) | .167 (.229) |
| 300° | 0 (.083) | 0 (.083) | 0 (.083) | 0 (0) |

Values are described as Median (IQR). FT: fitting task, MT: matching task, R-side: rotation side.

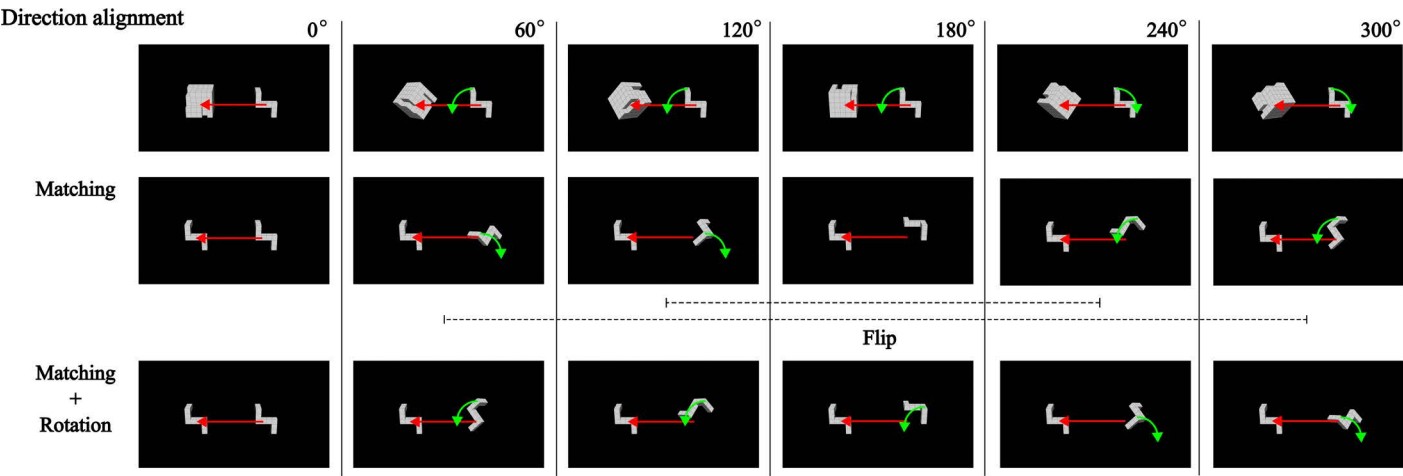

**Fig 8. Preprocessing of direction confounders.** Horizontal arrow in red: direction of matching, rotational arrow in green: direction of rotation. The first row displays the left rotation side in the fitting task. The second row shows the right rotation side in the matching task, controlling for the subjective direction. The third row further controls for the rotation direction by flipping the positions at 60° to 300° and 120° to 240° from the second row.

Task [$F(10, 290) = 7.180$, $p < .001$, $\eta_G^2 = .041$, $\eta_P^2 = .198$]. A simple effects analysis of the one-way RM-ANOVA for Task showed significant main effects of Task at five angles (all $p < .001$, $\eta_G^2 \geq .080$, $\eta_P^2 \geq .378$) except 180° ($\eta_G^2 = .015$, $\eta_P^2 = .052$). The post hoc pairwise comparison mostly showed significant differences between Task levels at five angles (all $p < .05$, $r^2 \geq .149$), except at 180° ($r^2 = .052$) and between FT and MT (flipped) at 300° ($r^2 = .078$). Thus, FT still took more time than MT at most angles from 0° (60°, 120°, 240°).

## 4. Discussion

This study examined how physical constraints influence cognitive strategies in FT under voluntary conditions by addressing the following three points:

(1) **Functional differences in cognitive strategies:** FT operated in a systematically directional manner in subjective reports and eye movements. This pattern differed markedly from MT (Figs 5A and 5B).

(2–1) **Behavioral asymmetry in FT:** The results revealed behavioral asymmetries in both FT and MT (Fig 6B), noted at a 120° rotation (Fig 6A). Interestingly, the asymmetries diminished for both tasks when collapsing rotation sides (Fig 6C).

(2–2) **Behavioral differences between FT versus MT:** Participants generally took longer to complete FT compared to MT ([Fig 7A]). However, when controlling for rotation and subjective comparison directions, the completion times for FT and MT were comparable at 180° ($r^2$ =.052) and 300° ($r^2$ =.078). Notably, among significant task differences, 60° ($r^2$ =.149) was distinctive compared to other angles ($r^2$ ≥.378, the least at 0°; [Fig 7B]). Echoing this, linearity explained 19% less variability in FT than MT ([Fig 6D]).

## 4.1. Directionality argument reflection

Subjective reports and eye-tracking data both support the directionality argument. Participants consistently showed a preference for matching/fitting smaller T-type objects with larger U-type objects, indicating a directional bias in cognitive strategies. This was further reinforced by eye-tracking data, which showed stronger dwell times on rotated U-type objects, underscoring a systematic directional bias in task processing. Together, these findings suggest that participants employed a directional strategy resembling the physical act of assembling jigsaw puzzles, where pieces are typically fit in a specific direction.

## 4.2. Detour argument reflection

Longer completion times, reduced linearity, and directionality in FT—mirroring physical jigsaw puzzles—support the detour argument. Nevertheless, unexpected breaches of symmetry and linearity in MT raise challenges for this perspective. Instead, behaviors observed in both tasks, after controlling for motion directions, suggest the involvement of both shortcuts and detours in FT. [Fig 9] illustrates this angle-dependent behavior, with shortcuts occurring at angles under 90° and detours at angles over 90°.

Angle-dependent behaviors themselves appear to hold. The face inversion effect [57,58] highlights orientation-sensitive cognitive thresholds at roughly ±90° [59–61], applicable to other stimuli by shaping prototypes or expertise in recognition [62–64]. Similarly, an investigation by Edelman and Bülthoff [65] implied a decline in object recognition accuracy for novel angles away from trained angles, mostly from 80° onward after repeated sessions. The role of object recognition in mental rotation may be complemented by physical rotation research by Gardony et al. [66], where participants appeared to make congruence judgments between two objects during same trials with angular disparities below 90°. The threshold concept extends to strategies; for example, occupational experiences of podiatrists might lead to strategic shifts from motor to non-motor imagery in mentally rotating inverted foot stimuli [67].

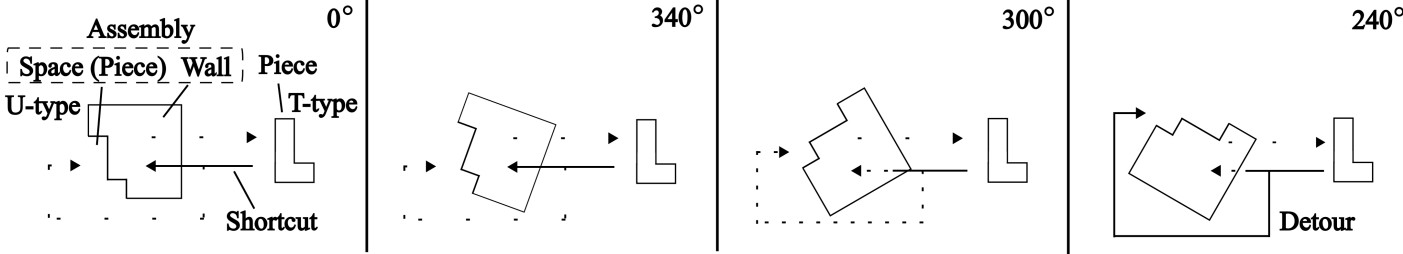

**Fig 9. Schematic of trajectories in mental jigsaw puzzles.** Dashed lines denote the degrees of represented routes. The present study refers to indented object as 'U-type' and protruding object as 'T-type.' The diagram is illustrative.

**Alternative explanations.** Given the voluntary nature of the conditions, several alternative explanations might account for the findings, though none fully explain them in isolation, particularly the systematic directionality and angle-dependent behavior. Instead, a combination of these explanations, along with the trajectories, may provide a more comprehensive understanding of the present findings. These include the following.

One possibility is detours in MT. While MT lacks physical barriers, the possibility of detours cannot be entirely dismissed. The mental process of matching objects in MT may be analogous to fitting T-type objects into U-type objects in FT. This process might involve mentally envisioning a tight-fitting 'box' around the T-type object, requiring detours to fit the T-type object within this U-type 'box.' This possibility is supported by the findings that linearity in MT accounted for only half the variability and by the observed behavioral asymmetry. However, MT explained 19% more variability than FT, suggesting that while detours may be involved in both tasks, they likely play a lesser role in MT.

A second potential explanation is coherence in concepts. Behavioral asymmetry may arise from conceptual coherence, akin to phenomena observed in the Stroop effect, Simon effect, or stimulus-response compatibility [68,69]. For example, Guiard [70] demonstrated how sensory laterality (ear stimulation) aligns with rotational directions in motor actions (steering wheel), illustrating an interaction between rotational direction and laterality. This concept could apply to the mental rotation tasks in this study. Specifically, the coherence between rotation [17,21] and translation (matching/fitting) directions could account for much of the observed behavioral asymmetry, as illustrated in Fig 10. When the translation direction is coherent with the object's rotational direction (e.g., a leftward translation paired with counterclockwise rotation), cognitive processing may be facilitated (e.g., leftness and leftness). Conversely, when translation and rotation directions are incoherent (e.g., leftward translation paired with clockwise rotation), cognitive processing may be delayed (e.g., leftness and rightness). This interpretation suggests that translation itself, regardless of whether it represents a shortcut or detour, can create systematic asymmetries due to the coherence of laterality. Such coherence in concepts may also influence behavioral asymmetries seen in motor responses [17,72], rotational motion aftereffects [73,74], laterality in both objects [75,76] and hands [25,77].

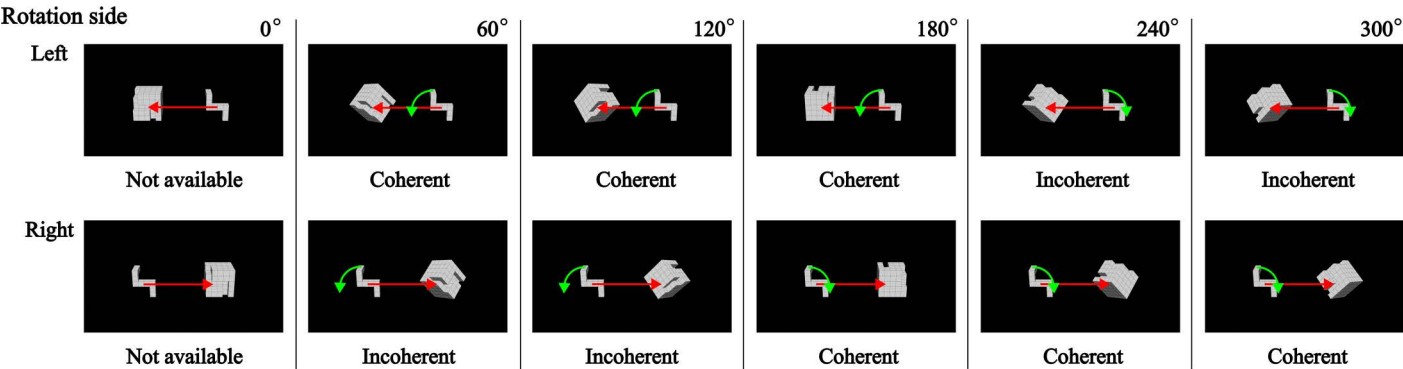

**Fig 10. Example combinations of directions in mental jigsaw puzzles.** Horizontal arrow in red: direction of translation, rotational arrow in green: direction of rotation. The head was selected as the starting point of rotation, given that this area may receive more biased fixation during the mental rotation of T-type objects [5,71]. Pairing a leftward comparison direction with a counterclockwise rotation from the head is coherent (leftness and leftness), compared to a leftward comparison with clockwise rotation from the head (leftness and rightness). This same asymmetry can be also achieved with the directions of translation and rotation switched. The same coherence behavior applies to object mental rotation tasks, achieved by visualizing T-type objects emerging from the blank spaces surrounding U-type objects. The present study refers to indented object as 'U-type' and protruding object as 'T-type'.

A third possibility is parts recognition. Hoffman and Richards [78] suggested that visual processing often involves segmenting shapes at their concave regions, which are perceived as natural dividing points. In this study, both T-type and U-type objects contained concave regions; however, U-type objects featured more extensive concave grooves. These more pronounced concave features in the U-type objects may require more complex cognitive processing as the visual system attempts to structurally segment these shapes. Such increased complexity could explain the longer reaction times observed in tasks involving the U-type objects, particularly at 0°, where no rotation is required.

The observed behavior at 0° may reflect object recognition or figure-ground segregation processes within the ventral pathway, including regions such as the lateral occipital complex, V2, and potentially V1 [79–82]. Conversely, the dorsal pathway, which is involved in mental rotation and includes the inferior and superior parietal cortices [27,83], may handle the mental manipulation of these objects. This division of labor between the ventral and dorsal pathways might explain the performance dip at a 60° angular disparity.

A fourth explanation is part/whole comparisons. Bilge and Taylor [16] demonstrated a tendency for bisected 3D objects to result in longer completion times than whole 3D stimuli, particularly at 0° and angles greater than 90°. This suggests that FT may employ a piecemeal strategy, whereas MT relies on a more holistic approach. The angle-dependent behavior observed in this study could result from a strategy shift occurring at 90°, as indicated in previous research [84].

A fifth potential explanation is motor planning. FT may require more complex motor planning than MT, given the physical nature of fitting objects. This aligns with research indicating longer execution times for more complex motor sequences [85,86]. While the actual translation time for detours may be short, the preparation time for movement could be longer, contributing to the extended completion times observed in FT.

Lastly, bodily constraints may also play a role. While this study attempted to minimize the beneficial effects of hand gestures [42], such restrictions may have inadvertently hindered performance. Previous research has shown reduced performance in awkward postures during hand laterality tasks, particularly when the hand is positioned on the back rather than the front [13,87], and this effect is influenced by right-handedness [88]. This is consistent with findings of slower responses in amputees with aesthetic prostheses [89] and expanded body representations in macaques after using tools [90,91]. These results suggest that even when the hand is resting in a natural position on the front, right-handedness and the resulting bodily constraints may have exerted a greater impact on FT under motor imagery compared to MT.

### 4.3. Comparison with relevant research

The relationship between jigsaw puzzle solving and mental rotation tasks has been explored in various contexts, including manual tasks. Aguilar Ramirez et al. [92] conducted an angle-free analysis and found that men tend to excel in 3D mental rotation tasks, while women outperform men in assembling two-dimensional (2D) jigsaw puzzles. The puzzle-assembling findings are consistent with earlier research on children using tablet-based 2D puzzles [93] but differ from research involving physical puzzles [94]. Aguilar Ramirez et al. [92] also performed a mediation analysis, interpreting their results as evidence of partial independence between visuospatial abilities in jigsaw puzzle solving and mental rotation tasks.

However, the analysis may not fully account for other potential factors. For example, sex differences in physical and motor characteristics [95], as well as differences in fine motor skills [94,96], may help explain the superior performance of females in assembling 2D jigsaw

puzzles. Another relevant factor is the frequent 90° flips involved in rotation during physical jigsaw puzzle-solving, where pieces are repeatedly flipped until they fit or match. This process may reflect a greater tendency among females to attempt fitting pieces into incorrect locations [92]. This flipping process is analogous to how Tetris® operates, with each press of the controller rotating a piece by 90°. As such, a trial-and-error approach may play a more crucial role than visuospatial abilities in this context. Previous research on the cognitive training effects of Tetris® found no significant cognitive transfer, such as mental rotation performance, but rather task-specific transfer, rhetorically referring to the training as 'Game over' [97]. Nevertheless, mental rotation itself may still contribute to performance in the block game [3].

Interestingly, this study's secondary analysis found no gender differences in eye-tracking or behavioral outcomes for both FT and MT [46], which used a pair of 3D T-type objects from Shepard and Metzler [14], alongside a pair of 3D smaller T-type objects and their corresponding larger U-type objects. Hence, the present research provides a more direct, 'apple-to-apple' comparison, investigated through detailed angle-specific analysis in both object mental rotation and mental jigsaw puzzle tasks [7], thereby facilitating a deeper understanding of functional differences in visuospatial strategies.

Other research explored cognitive tasks using jigsaw puzzle pieces [4] or ghost figures [6,15], both in 2D formats. These studies share similarities with the present research in that mental jigsaw puzzles appear to engage mental rotation. Notably, monotonic, linearity-like behaviors have been observed [6,15], and increased neural activation in the dorsolateral prefrontal cortex at larger rotation angles has been documented [4], aligning with findings from a meta-analysis on mental rotation [27].

In terms of shapes, both piece and asembly pairs add up to primitive shapes, such as square-like shapes [4] or circles [6], resembling the cube used in the present study. Regarding individual assembly shapes, Mutlu et al. [4] employed white puzzle pieces fully framed by dark edges and gray assemblies, while Frick et al. [15] and Frick and Pichelmann [6] employed light-gray ghost figures fully framed by dark edges and dark circles or squares. Both studies employed T-type puzzle pieces that were color-matched with the corresponding T-type spaces of the U-type assemblies. Hence, despite the U-type and T-type pairings, these visual cues could imply a predominantly T-type to T-type pairing. Indeed, Frick and Pichelmann [6] employed a factor analysis to demonstrate that the ghost puzzle tasks measured the same ability as established object mental rotation tasks. This reflects the interchangeable use of 'fit' and 'match' in these studies [4,6,15]. The presence of fully framed U-type figures may have provided the inner space with a clear meaning, guiding the participants' perception of the T-type shapes.

In contrast, the present study's fitting areas for U-type pieces are not fully framed by unit cubes, resulting in partial shapes. This is similar to the process in physical jigsaw puzzles, where fitting mostly involves adding pieces along partially shaped edges. Notably, even with incomplete information, a whole can be sometimes effortlessly perceived. Specifically, figures can still be recognized even when partially shaped, as illustrated by the Rubin vase, Kanizsa triangle, and Gregory's Dalmatian dog illusions. Once the figure is recognized, its identification becomes easier in subsequent trials. This recognition process could be facilitated in FT by identifying the space within the U-type assembly and matching the two T-type pieces, typically using shortcuts. Hence, those with stronger Gestalt perception skills—typically, the ability to form a meaningful whole from parts—may have relied on such shortcuts, whereas participants who found it more challenging to fully recognize the figure might have employed alternative strategies, such as detour strategies via motor imagery.

Previous mental rotation studies using fMRI have indicated that object-related affordances influence activation in motor-related regions. For instance, Vingerhoets et al. [28] required participants to compare two objects—either hands or tools—and determine whether they were the same or different. Their imaging data revealed activation in primary regions associated with mental rotation, including the superior parietal lobule, for both hand and tool stimuli. However, they observed a key difference in premotor activation: hand stimuli elicited bilateral premotor activation, while tool stimuli primarily activated the left premotor cortex in right-handed individuals. These findings were interpreted as evidence that the afforded actions of objects play a role in engaging motor-related regions through motor imagery in a way that reflects natural, real-world interactions [28]. In the context of the present study, puzzle pieces may similarly elicit affordance-driven activation patterns in motor-related regions, potentially clarifying participants' trajectory strategies.

### 4.4. Limitations

The sample size in this study was determined based on a behavioral pilot study using unique methodologies [7]. While this provided a foundation, the calculation may not fully account for all analyses conducted in this study, potentially leading to underestimation or overestimation in certain areas. To mitigate this, effect sizes have been provided where relevant, and raw data are extensively available. Another limitation is the potential reduction in internal validity due to the nature of voluntary conditions in the study design, as discussed in the alternative explanations for the detour argument. Participants were allowed to choose their cognitive strategies, which introduced variability that complicates the control of confounders, such as directionality. However, this variability mirrors real-world scenarios, providing valuable insights into natural cognitive processes.

Other limitation pertains to the angular disparities employed in this study. Angles of 0°, 60°, 120°, 180°, 240°, and 300° were systematically used to assess rotation and translation strategies. While this approach ensures consistent measurement and comparability, it may not fully capture behaviors associated with more randomized angular variations, which could better simulate real-world scenarios. One possible improvement could involve the use of angular variation bins, as employed by Gardony et al. [66].

Additionally, this study exclusively tested configurations involving smaller T-type objects and larger U-type objects, referencing the traditional stimuli by Shepard and Metzler [14], but did not examine cases with smaller U-type objects and larger T-type objects. Under a piece-to-assembly perspective, it is expected that smaller U-type objects could also move toward larger T-type objects, as seen when plugging a U-type protective cap onto a T-type terminal. However, this perspective may vary depending on context, and future research could investigate whether this reversed configuration elicits similar cognitive strategies or introduces distinct spatial biases.

### 4.5. Future studies

Building on the findings and discussion points of this study, future research could explore several avenues for further investigation.

First, future experiments could examine the neural correlates of translation strategies and object recognition, potentially using fMRI or EEG, to clarify the underlying cognitive mechanisms. For instance, investigating the activation of motor-related regions, as highlighted by Vingerhoets et al. [28], could provide deeper insights into the role of motor imagery and affordances during mental jigsaw puzzle tasks.

Second, the role of conceptual coherence warrants further exploration. The behavioral asymmetry observed in this study may originate from conceptual coherence, where the output

of mental rotation is influenced by decision-making steps, disrupting the expected pure symmetry and linearity across angles.

Third, more detailed eye-tracking analyses could be conducted. The current research primarily used two bisectional sides to investigate directional bias. However, future studies could analyze fixation patterns on specific parts of the objects to gain valuable insights into strategies such as piecemeal versus holistic approaches [16]. This could offer a deeper understanding of cognitive processing during FT and MT.

Fourth, metacognitive approaches [98] could be explored in addition to objective data to further investigate the role of detours. Ongoing research aims to examine participants' self-awareness of their strategies during MT and FT. Understanding how individuals reflect on their cognitive strategies could provide valuable insights into these tasks.

Lastly, incorporating hand tracking in future research could open new avenues for understanding the role of motor function and its interaction with visuospatial strategies. Upcoming research aims to investigate how hand-tracking data could enhance our understanding of trajectory strategies during FT and MT. This approach could bridge the gap between the start and end points in cognitive tasks by examining the middle stages, offering a more comprehensive view of how physical constraints influence cognitive strategies.

## 5. Theoretical contributions and concluding remarks

This research demonstrates that participants approached mental jigsaw puzzles using strategies similar to those employed in physical puzzle-solving, with a notable emphasis on systematic directionality under voluntary conditions. The study introduces mental jigsaw puzzles as a pioneering experimental methodology that integrates mental rotation and translation. Although behavioral asymmetry, reduced linearity, and angle-dependent behavior were observed, the roles of detours remain partially unclear, underscoring the need for further research.

## Acknowledgments

The authors thank Dr. Kurt Debono, a former laboratory member who opted for anonymity, and Yana Yu for their support of the eye-tracking system.

## Author contributions

**Conceptualization:** Tsuyoshi Yoshioka.

**Data curation:** Tsuyoshi Yoshioka.

**Formal analysis:** Tsuyoshi Yoshioka.

**Funding acquisition:** Jun Saiki.

**Investigation:** Tsuyoshi Yoshioka.

**Methodology:** Tsuyoshi Yoshioka, Hiroyuki Muto, Jun Saiki.

**Project administration:** Tsuyoshi Yoshioka.

**Resources:** Jun Saiki, Tsuyoshi Yoshioka.

**Software:** Tsuyoshi Yoshioka.

**Supervision:** Jun Saiki, Hiroyuki Muto.

**Validation:** Tsuyoshi Yoshioka.

**Visualization:** Tsuyoshi Yoshioka.

**Writing – original draft:** Tsuyoshi Yoshioka.

**Writing – review & editing:** Tsuyoshi Yoshioka, Jun Saiki, Hiroyuki Muto.

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
