## [Decision Letter · Decision Letter 0]

23 Jan 2025

Dear Dr. Yoshioka,

Thank you for submitting your manuscript to PLOS ONE. After careful consideration, we feel that it has merit but does not fully meet PLOS ONE’s publication criteria as it currently stands. Therefore, we invite you to submit a revised version of the manuscript that addresses the points raised during the review process.

We look forward to receiving your revised manuscript.

Kind regards,

Sheikh Arslan Sehgal, PhD

Academic Editor

PLOS ONE

Reviewers' comments:

Reviewer's Responses to Questions

**Comments to the Author**

1. Is the manuscript technically sound, and do the data support the conclusions?

Reviewer #1: Yes

Reviewer #2: Partly

2. Has the statistical analysis been performed appropriately and rigorously?

Reviewer #1: Yes

Reviewer #2: Yes

3. Have the authors made all data underlying the findings in their manuscript fully available?

Reviewer #1: Yes

Reviewer #2: No

4. Is the manuscript presented in an intelligible fashion and written in standard English?

Reviewer #1: Yes

Reviewer #2: Yes

Reviewer #1: 1) The originality and novelty of the paper should be detailed in a separate paragraph/sub-section.

2) Please discuss the future studies and study limitations in detail.

3) Figures 6 and 7 are unclear. Kindly replace them with clearer versions for better understanding.

4) There are grammar, spelling, and punctuation errors throughout the manuscript, which would need to be addressed.

Reviewer #2: General Comment:

This study looked at how people think when solving mental jigsaw puzzles. It combined two ideas: mental rotation (how we imagine turning objects in our minds) and translation (how we move them), focusing on how direction affects problem-solving.

There are key findings found in the paper: (1) Physical Constraints: Unlike regular mental rotation tasks, these puzzles had physical limits that changed how people looked at them (2) Eye Movements: The study found that when solving these puzzles, people tended to look at smaller male objects and direct them toward larger female objects (3) Completion Times: People took longer to finish the puzzles, and their problem-solving paths were less straightforward, similar to how they solve physical puzzles. (4) Behavioral Patterns: The way people approached these puzzles showed patterns similar to those seen in regular mental rotation tasks.

Although the study compared different ways of thinking and found some interesting similarities, it didn't completely explain how detours (taking longer paths to solve a puzzle) worked. This suggests that more research is needed to understand this better.

Minor Review:

1. In the acknowledgment, the authors mentioned an anonymous contributor; in my opinion, it must be stated clearly, who is an anonymous contributor. At least there is a general explanation about him or her or them.

2. Section 2.3 mentioned “The experiment was conducted in a dark room using an EyeLink® 1000 Plus Desktop …”. Please explain what kind of darkroom was used, how dark it is, etc.

3. The connection between the text in the paper and the images, at the end of the paper, should be related with clear numbering. For example, section 3.1 mentioned, “DTR. Fig 5a illustrates task…” We cannot find easily which figure is Figure 5.

4. Section 2.5 mentioned about FIT as Fitting Task. It should be FT, if I am not mistaken.

5. Section 4.3 mentioned about fRMI. It should be explained.

Major Review:

1. Several figures in the paper mentioned information about Male and Female, but it is not described in the paper. For example in Figure 7. It must be explained in the text.

2. Section 2.1 mentions that there are 40 participants. Mentioned the characteristics of these participants. Are they experts? Common human? Students? Lecture? Etc.

3. Is gender issue part of this research? If yes please give an additional explanation in the paper for each section. For example in section 2.6, no explanation about male and female data (Behavioral and eye tracking data)

4. I have difficulties finding an explanation of the Translation issue mentioned in the abstract. Should be justified.

5. Section 2.4 mentioned, “… A total of 576 stimuli were randomly assigned to each subblock …” Is the data open, and what kind of stimuli were used in the experiment?

6. Section 2.5 mentions the angles employed in the research are Angle: 0°, 60°, 120°, 180°, 240°, 300°. How to measure this angle? How about another angle or even a random angle? It can make the experiment more realistic.

Comment to the author:

Congratulations, the article is easy to read and understand. The problem is fundamental, but the explanation is complete and scientific. A revision is required to make the paper more meaningful and increase its scientific weight.

**Do you want your identity to be public for this peer review?** For information about this choice, including consent withdrawal, please see our Privacy Policy

Reviewer #1: No

Reviewer #2: **Yes: ** Setiawan Hadi

---

## [Author Response · Author response to Decision Letter 1]

6 Feb 2025

February 6, 2025

Manuscript ID: PONE-D-24-50498

Manuscript Title: Functional perspectives in mental jigsaw puzzles: Insights from eye-tracking, questionnaire, and behavioral data

Academic Editor

PLOS ONE

Dear Sheikh Arslan Sehgal, PhD,

We thank you and the reviewers for the constructive feedback on our manuscript. Below, we provide a point-by-point response to all comments and outline the revisions made to address them. We hope the revised manuscript addresses all concerns and enhances its clarity and scientific contribution.

Reviewer #1

1) The originality and novelty of the paper should be detailed in a separate paragraph/sub-section.

Response:

Thank you for your valuable suggestion. We recognize that the previous version did not explicitly emphasize the originality and novelty of this study. In response, we have removed the separate subsection and instead integrated a more explicit discussion of the novelty and originality within the introduction to ensure it is seamlessly incorporated into the broader context of the study.

To address this, we have made the following key revisions in Section 1 (Introduction), across paragraphs 3 to 5 on pages 4 and 5:

• We highlight how this study bridges the gap between mental rotation tasks and real-world spatial problem-solving by introducing a task that integrates both rotation and translation under physical constraints.

• We clarify how this study extends prior research on puzzle-like tasks, many of which were 2D-based or lacked interlocking constraints. By incorporating interlocking mechanisms, our research provides new insights into the role of physical constraints in cognitive strategies.

• We present mental jigsaw puzzles as an experimental framework that captures the interaction between mental rotation and real-world spatial reasoning—a perspective not systematically explored in previous research.

These updates ensure that the novelty and originality of the study are clearly presented within the introduction.

2) Please discuss the future studies and study limitations in detail.

Response:

We have expanded both the limitations and future studies sections to provide a comprehensive discussion:

• Study Limitations:

o Addressed limitations related to the fixed angular disparities (0°, 60°, 120°, 180°, 240°, 300°), suggesting that randomized angles could improve real-world scenarios.

o Highlighted the variability introduced by voluntary cognitive strategies as both a limitation and insight into real-world processes.

o Acknowledged the reliance on reaction time as a proxy for detour behavior, recommending complementary neural and behavioral measures for future research.

o Clarified that this study focused on configurations with smaller protruding objects and larger indented objects, without examining cases involving smaller indented objects and larger protruding objects

• Future Studies:

o Proposed exploring neural correlates of translation and detour strategies using fMRI or EEG, including object recognition and motor imagery mechanisms.

o Suggested investigations into conceptual coherence influencing behavioral asymmetry during mental rotation tasks.

o Recommended detailed eye-tracking analyses to distinguish piecemeal versus holistic strategies in FT and MT.

o Highlighted the value of metacognitive approaches to study participants' self-awareness of strategies.

o Emphasized incorporating hand tracking to explore trajectory planning in cognitive tasks.

These revisions are included in Sections 4.4 (Limitations) and 4.5 (Future studies) on pages 34 to 36.

3) Figures 6 and 7 are unclear. Kindly replace them with clearer versions for better understanding.

Response:

Thank you for your feedback. We have updated Figures 6 and 7 to improve clarity and readability. Additionally, we have refined their captions for better comprehension.

• Figure 6:

o Improved layout and axis labeling for enhanced readability.

o Added color shading in panel D to illustrate confidence intervals more effectively.

o Removed object gender categories to avoid potential confusion.

• Figure 7:

o Optimized layout and increased font size in panel B for better visibility.

o Updated captions to provide more detailed explanations.

o Removed object gender categories for brevity.

We believe these revisions improve the clarity of the figures and align them more closely with the manuscript’s content. Please let us know if further adjustments are needed.

4) There are grammar, spelling, and punctuation errors throughout the manuscript, which would need to be addressed.

Response:

We conducted a thorough proofreading process to address grammar, spelling, and punctuation issues. Specifically:

• Ensured consistency and accuracy in grammar and punctuation.

• Corrected typographical errors.

• Improved sentence structure for clarity and readability.

A track-changed version of the manuscript highlights these revisions.

Reviewer #2

General Comment:

Thank you for your insightful summary and feedback on our study. Below, we address the specific comments provided.

Minor Review:

1) In the acknowledgment, the authors mentioned an anonymous contributor; it must be stated clearly who they are.

Response:

Thank you for your comment. We have clarified in the revised Acknowledgments section on page 37 that the individual was a former laboratory member who explicitly requested anonymity. This revision ensures transparency while respecting their preference.

2) Section 2.3 mentions “The experiment was conducted in a dark room using an EyeLink® 1000 Plus Desktop…”. Please explain the kind of darkroom used.

Response:

We expanded Section 2.3 (Apparatus) on page 13 to clarify that the experiment was conducted in a private dark room with no ambient lighting. While not entirely light-sealed, the primary light source was the monitor, ensuring distraction-lessened conditions.

3) The connection between the text and the figures should be clearer. For example, section 3.1 mentions “DTR. Fig 5a illustrates task…” but it is difficult to locate the figure.

Response:

Thank you for pointing out the difficulty in locating figures. We recognize the importance of ensuring clear connections between the text and corresponding figures.

To improve clarity, we have made the following adjustments:

1. Numbering alignment: We have ensured that figure references in the text match the numbering format used within the figures themselves. For example, Fig 5A in the text corresponds to the labeled A in Fig 5.

2. Omission of "Note": The word "Note" has been removed from all figure captions to improve readability and avoid confusion regarding whether it belongs to the caption or the main text.

3. Figure legend placement: The figure legends have been repositioned next to the figure titles for better readability and consistency.

4. Journal formatting compliance: Figure references have been maintained in the required format (e.g., Fig 5A, Fig 5B).

Additionally, we acknowledge that the search ("find") function may not detect attached figures due to the journal-mandated naming convention (e.g., Fig5.tif instead of Fig 5.tif). Unfortunately, modifying this convention is not permitted under the journal’s guidelines. As a workaround, we suggest searching for figure references without a space (e.g., searching for "Fig5" instead of "Fig 5") to locate figures efficiently.

We hope these revisions enhance figure accessibility and clarity. Please let us know if further improvements are needed.

4) Section 2.5 mentions "FIT" instead of "FT".

Response:

Thank you for pointing this out. Upon review, we noticed that the term "FIT" does not appear in the manuscript. However, we will ensure that "FT" (Fitting Task) is used consistently throughout the manuscript.

5) Section 4.3 mentions "fRMI". It should be explained.

Response:

Thank you for your comment. Upon review, we noticed that the term "fRMI" does not appear in the manuscript and believe this may have been a typographical error. We assume "fMRI" (functional magnetic resonance imaging) was referred to. To clarify, we have expanded Section 4.3 (Discussion), paragraph 7, on pages 33 to 34 to explain the fMRI findings from Vingerhoets et al. (2002). fMRI measures brain activity by detecting changes in oxygenation levels in blood flow using magnetic waves. In the referenced study, this technology was employed to investigate how object-related affordances, such as tool and hand stimuli, influence motor-related regions of the brain during mental rotation tasks.

Major Review:

1) Figures include references to "male" and "female" objects but lack explanation.

Response:

Thank you for your comment regarding the terminology used for "male" and "female" objects, as commonly seen in electronics to describe connector types. We recognize that these terms may cause confusion, particularly because they are also associated with human gender.

To enhance clarity and avoid ambiguity, we have replaced "male" and "female" with terminology that aligns more closely with both jigsaw puzzle contexts and connector shapes, as introduced in the manuscript:

• U-type (indented shapes)

• T-type (protruding shapes)

These revised terms help to prevent confusion while maintaining consistency with the study’s task descriptions.

Additionally, we have updated the figure captions and legends in Figures 1, 3, 5, 6, 7, and 9 to reflect these terminology changes, ensuring clarity and consistency throughout the manuscript.

We believe these revisions effectively address the concern and eliminate potential confusion. Please let us know if further refinements are required.

2) Section 2.1 mentions 40 participants. Please describe their characteristics.

Response:

We clarified in Section 2.1 (Participants) on page 11 that participants were undergraduate and graduate students, primarily right-handed, as determined using the FLANDERS Handedness Questionnaire.

3) Is gender a factor in this research?

Response:

No, gender is not a primary factor in this research. However, secondary analyses were conducted, and the results are briefly reported. No significant gender differences were found in either behavioral or eye-tracking indices. This is described in Section 2.6 (Data curation), Paragraph 3, on page 16, and in Section 4.3 (Comparison with relevant research), Paragraph 3, on page 32.

We acknowledge that the use of "female" and "male" to describe object characteristics, following connector-type terminology, may have caused confusion, as these terms are also associated with human gender. To address this, we have revised the terminology throughout the manuscript, as detailed in (1) of the Major Review, to avoid ambiguity. This clarification has been made more explicit in the revised manuscript to ensure transparency.

4) The "Translation issue" mentioned in the abstract should be clarified.

Response:

Thank you for your comment regarding the need to clarify the translation issue in the study. We have revised the manuscript to provide a more explicit discussion of translation and its relevance to the findings, particularly in the second alternative explanation within Section 4.2 (Detour argument reflection), Paragraph 5, on page 28.

1. Explicit integration of Fig 10 into the main text:

o The content previously included in the Fig 10 caption has been expanded and incorporated into the main text.

o We now provide a step-by-step explanation of how coherence between rotation and translation directions may either facilitate or delay cognitive processing (e.g., "leftness and leftness" vs. "leftness and rightness").

2. Expanded discussion:

o We clarified that conceptual coherence in laterality between rotation and translation directions may contribute to systematic asymmetries, regardless of whether translation represents a shortcut or a detour.

These revisions ensure that the translation issue is clearly articulated while also emphasizing the need for further investigation into the role of translation in mental jigsaw puzzles. We believe these updates provide a clearer and more structured response to the feedback. Please let us know if further refinements are needed.

5) Section 2.4 mentions, “A total of 576 stimuli were randomly assigned…” Is the data open, and what stimuli were used?

Response:

Yes, the data is publicly available at the OSF link: https://osf.io/4ku38/. We used the stimuli consisting of either a pair of smaller protruding objects or a pair comprising a smaller protruding object and a larger indented object. Example stimuli are detailed in Fig 1.

6) Section 2.5 mentions angles (0°, 60°, 120°, 180°, 240°, 300°). How are these measured, and why not random angles?

Response:

We clarified the method of angle measurement in Section 2.2 (Stimuli) on page 12, explaining how these angles were systematically applied to ensure consistency across trials. Additionally, we addressed the limitation of using fixed angles instead of randomized variations in Section 4.4 (Limitations), Paragraph 2, on page 35. We acknowledged that using randomized angles could better enhance real-world scenarios.

Comment to the Author:

Thank you for your positive feedback. We appreciate your recognition of the clarity and scientific merit of the manuscript and have revised the paper to address all concerns.

We hope these revisions meet the reviewers' expectations. Please let us know if further clarification is needed.

Sincerely,

Tsuyoshi Yoshioka

Graduate School of Human and Environmental Studies, Kyoto University

Email: yoshioka.tsuyoshi.yt@gmail.com

---

## [Decision Letter · Decision Letter 1]

3 Mar 2025

Functional perspectives in mental jigsaw puzzles: Insights from eye-tracking, questionnaire, and behavioral data

PONE-D-24-50498R1

Dear Dr. Yoshioka,

We’re pleased to inform you that your manuscript has been judged scientifically suitable for publication and will be formally accepted for publication once it meets all outstanding technical requirements.

Kind regards,

Sheikh Arslan Sehgal, PhD

Academic Editor

PLOS ONE

Additional Editor Comments (optional):

Reviewers' comments:

Reviewer's Responses to Questions

**Comments to the Author**

Reviewer #2: All comments have been addressed

2. Is the manuscript technically sound, and do the data support the conclusions?

Reviewer #2: Yes

3. Has the statistical analysis been performed appropriately and rigorously?

Reviewer #2: Yes

4. Have the authors made all data underlying the findings in their manuscript fully available?

Reviewer #2: Yes

5. Is the manuscript presented in an intelligible fashion and written in standard English?

Reviewer #2: Yes

Reviewer #2: The study effectively integrates mental rotation and translation in the context of mental jigsaw puzzles, and the inclusion of eye-tracking data, questionnaires, and behavioral data provides a comprehensive approach to understanding cognitive strategies.

**Do you want your identity to be public for this peer review?** For information about this choice, including consent withdrawal, please see our Privacy Policy

Reviewer #2: **Yes: ** Setiawan Hadi

---

## [Editor Report · Acceptance letter]

PONE-D-24-50498R1

PLOS ONE

Dear Dr. Yoshioka,

I'm pleased to inform you that your manuscript has been deemed suitable for publication in PLOS ONE. Congratulations! Your manuscript is now being handed over to our production team.

Kind regards,

on behalf of

Dr Sheikh Arslan Sehgal

Academic Editor

PLOS ONE